# Rhythmicity is linked to expression cost at the protein level but to expression precision at the mRNA level

David Laloum [1,2], Marc Robinson-Rechavi [1,2] *

1 Department of Ecology and Evolution, Batiment Biophore, Quartier UNIL-Sorge, Université de Lausanne, Lausanne, Switzerland, 2 Swiss Institute of Bioinformatics, Batiment Génopode, Quartier UNIL-Sorge, Université de Lausanne, Lausanne, Switzerland

* marc.robinson-rechavi@unil.ch

**Data Availability Statement:** Data and scripts are available at: https://github.com/laloumdav/cost_noise_conservation_rhythmicity.

## Abstract

Many genes have nycthemeral rhythms of expression, *i.e.* a 24-hours periodic variation, at either mRNA or protein level or both, and most rhythmic genes are tissue-specific. Here, we investigate and discuss the evolutionary origins of rhythms in gene expression. Our results suggest that rhythmicity of protein expression could have been favored by selection to minimize costs. Trends are consistent in bacteria, plants and animals, and are also supported by tissue-specific patterns in mouse. Unlike for protein level, cost cannot explain rhythm at the RNA level. We suggest that instead it allows to periodically reduce expression noise. Noise control had the strongest support in mouse, with limited evidence in other species. We have also found that genes under stronger purifying selection are rhythmically expressed at the mRNA level, and we propose that this is because they are noise sensitive genes. Finally, the adaptive role of rhythmic expression is supported by rhythmic genes being highly expressed yet tissue-specific. This provides a good evolutionary explanation for the observation that nycthemeral rhythms are often tissue-specific.

## Author summary

For many genes, their expression, *i.e.* the production of RNA and proteins, is rhythmic with a 24-hour period. Here, we study and discuss the evolutionary origins of these rhythms. Our analyses of data from different species suggest that the rhythmicity of protein level may have been favored by selection for cost minimization. Furthermore, we have shown that cost cannot explain the rhythmic variations in RNA levels. Instead, we suggest that it periodically reduces the stochasticity of gene expression. We also found that genes under stronger purifying selection are rhythmically expressed at the mRNA level, and propose that this is because they are noise-sensitive genes. Finally, rhythmic expression involves genes that are often highly expressed and tissue-specific. This provides a good evolutionary explanation for the tissue-specificity of these rhythms.

**Funding:** MRR received grant 31003A_173048 of Schweizerischer Nationalfonds zur Förderung der Wissenschaftlichen Forschung (SNF) https://www.snf.ch/en The funders had no role in study design, data collection and analysis, decision to publish, or preparation of the manuscript.

**Competing interests:** The authors have declared that no competing interests exist.

## Introduction

Living organisms have to adapt to complex and changing environments. Physiological systems able to accommodate themselves to changing but predictable circumstances are expected to have a higher stability of survival and reproduction [1]. The strongest predictable change for most organisms is the light/dark cycle and the associated nycthemeral temperature variations. In an organism, "circadian rhythms" denote entities characterized by an endogenous and entrainable oscillator clock which is able to persist in constant conditions (such as in constant darkness) and whose phases can be altered (reset or entrained). However, many physiological systems display non-autonomous nycthemeral rhythms, directly or indirectly controlled by the local clock or mainly by the environment itself, or by both [2–6]. Such rhythms are found at all levels: molecular, cellular, organs, and behavioural, and several regulatory networks appear to play roles in the synchronization of these levels [7].

Here, we studied some of the evolutionary costs and benefits that shape the rhythmic nature of gene expression at the RNA and protein levels. For this, we analysed several characteristics that we expect to determine the rhythmic nature of gene expression in a the trade-off between its advantages (economy of energetic costs over 24h, ribosomal non-occupancy) and disadvantages (costs of complexity due to precise temporal regulation). We have not studied here the potential negative effect of proteins when they are not needed, such as mis-interaction of proteins, or unwanted enzymatic activity. We call "rhythmic genes" all genes displaying a 24-hours periodic variation of their mRNA or protein level, or of both, constituting the nycthemeral transcriptome or proteome. Their rhythmic expression can be entrained directly by the internal clock but also directly or indirectly by external inputs, such as the light-dark cycle or food-intake [2–6]. Hence we use the term "nycthemeral" to avoid confusion with the specific features of "circadian" rhythms, although these are included in the nycthemeral rhythms, which includes downstream regulated genes. Because the alternation of light and dark can be considered as a permanent signal for most life on earth [8], we consider that the entirety of nycthemeral biological rhythms are relevant as a phenotype under selection.

### Rhythmic gene expression: An adaptation in cycling environments

The endogenous nature of circadian rhythms is a strategy of anticipation. The ability to anticipate improves the adaptation of organisms to their fluctuating environment. Most rhythmic genes are tissue-specific [9–11], *i.e.* a given gene can be rhythmic in some tissues, and constantly or not expressed in others, which means that their rhythmic regulation is not a general property of the gene and is therefore expected to be advantageous only in those tissues in which they are found rhythmic. This argues that rhythmic regulation has costs, since it is not general. These costs are probably related to the complexity of regulation to maintain precise temporal organisation, and could also be related to toxicity. Thus, cyclic biological systems are expected to have adaptive origins. Gene expression is costly for the cell in terms of energy and cellular materials usage. Wang et al. [12] have shown that in the liver of mice, abundant proteins that are required at one time are down-regulated at other times, apparently to economize on overall production. Wang et al. also reported that at each time-point, the total metabolic cost was 4 fold higher for the set of cycling genes compared to the non-cycling genes set at both transcriptional and translational levels [12]—although the proteomic data used from mouse fibroblasts appear to have been underestimated and have since been corrected [13]. Furthermore, we have shown in our previous work that rhythmic genes are largely enriched in highly expressed genes and that the differences in rhythm detection obtained between highly and lowly expressed genes either reflect true biology or a lower signal to noise ratio in lowly expressed genes [14]. Here, we present results supporting the hypothesis that cyclic expression

of highly expressed proteins were selected as part of a "low-cost" strategy to minimize the overall use of cellular energy. Thus, a first evolutionary advantage given by rhythmic biological processes would be a reduction of the overall cost (over a 24-hour period), compared to the costs generated over the same period by a sufficient constant level of proteins. Furthermore, we provide a first explanation for the tissue-specificity of rhythms in gene expression by showing that genes are more likely to be rhythmic in tissues where they are specifically highly expressed.

## Noise and cost reduction

Expression costs at the protein level are at least 160 times higher than at the RNA level (section 3 in S1 File). Thus, it requires considerably more metabolic activity to produce a significant change in protein than in transcript levels. Moreover, small increases in protein expression levels have been shown to incur energy costs large enough to be opposed by natural selection, at least in bacteria [15, 16]. Consequently, protein synthesis costs are more constraining than RNA synthesis costs, especially in eukaryotes for which translation is the major contribution to making expression costs visible to selection [16].

Many transcripts show nycthemeral fluctuations without rhythmicity of their protein abundances, even when measured in the same study. This is the case for instance in mouse liver [17] or in plants [18]. Yet relative to protein synthesis costs, costs at RNA level are probably too small to provide a satisfactory evolutionary explanation of rhythms at mRNA level.

Thattai et al. [19] and Hausser et al. [20] propose that living systems have made tradeoffs between energy efficiency and noise reduction. Indeed, the control of noise (stability against fluctuations) plays a key role in the function of biological systems (*i.e.* in the robustness of gene expression) as supported by observations of a noise reduction during key periods, e.g. in development [19, 21]. These considerations lead to predictions which we study here: i) a strategy to periodically decrease stochasticity for genes with rhythmically accumulated mRNAs; ii) a cost-saving strategy for genes whose protein expression is rhythmic; and iii) a combined strategy for genes rhythmic at both levels.

## Results

### Cyclicality of costliest genes

Expression costs at the protein level are at least 160 times higher than at the RNA level (section 3 in S1 File). They are dominated by the cost of translation [15, 16]. This is mainly due to the higher abundance of proteins relative to transcripts, by a factor of 1000 (section 3 in S1 File). Thus, based on the calculation developed by Lynch and Marinov [16], we estimated the expression cost ($C_p$) of each gene by the formula (*1*) which takes into account the averaged amino-acid (AA) synthesis cost ($\bar{c}_{AA}$) of the protein (S1 Table), protein length ($L_p$), and protein abundance ($N_p$).

$$C_p = N_p L_p \bar{c}_{AA} \tag{1}$$

Other costs such as AA polymerization or protein decay are not expected to change our results, since longer or higher expressed proteins need more chain elongation activity and degradation (section 4 in S1 File). We show in S1 File (section 4) that the costs of maintaining proteins is higher for highly expressed than for lowly expressed proteins, even taking into account that highly expressed proteins have longer half-lives (also see Discussion). Thus, the cost provided by formula (*1*) is representative of gene expression costs and can be used for comparison between genes. To compare expression costs between rhythmic and non-rhythmic proteins, we calculated the average and the maximum protein expression level over time-points (see

Methods, Eqs (2) and (3)) (Fig 1d). The AA biosynthesis costs estimated in *E. coli* (S1 Table) were used as representative for all species since biosynthetic pathways are nearly universally conserved [15].

If the biological function of a protein is periodic, we expect the rhythmic regulation of its expression to be determined by a trade-off between the benefits of not producing proteins when they are not needed (energy costs saved, as we did not include the potential toxicity of unneeded proteins) and the potential costs involved in making it rhythmic and temporally coordinated (costs of complexity) (Fig 1b).

This leads to expect costlier genes to be more frequently rhythmic. First, we confirm that cycling genes are indeed enriched in highly expressed genes, as previously reported [12, 14]. The higher cost of cycling genes (Fig 2a and S2 Fig) was especially correlated with their higher expression levels, observed both at the protein level (Fig 2d and 2e) and at the RNA level (S1 Fig). We find rhythmic proteins to be longer only in mouse and in cyanobacteria (Fig 2c). However, conserved genes are known to be longer [22, 23] and are often more expressed [24, 25], therefore more expensive to produce, which could explain the rhythmic expression we observe for longer proteins. On the other hand, the higher cost for rhythmic proteins in *Synechococcus elongatus* appears only due to their length. Finally, in mouse liver, cycling proteins seem to contain more expensive amino acids than non-rhythmic proteins (Fig 2b).

Overall, results are consistent with the expectation that costlier genes are preferentially under rhythmic regulation. Furthermore, our results show that while rhythmicity at protein level lowers costs, it does not drive the costs of rhythmic genes down to the level of constant proteins. A more specific prediction from this hypothesis is that each gene should be found rhythmic specifically in tissues where a high expression of that gene is needed, *i.e.* where its function is costlier.

## Genes tend to be rhythmic in the tissues in which they are highly expressed

To test this hypothesis, we used a mouse circadian dataset with 11 tissues (transcriptomics from Zhang et al. [9], S2 Table). For each gene we separated the tissues into two groups, those for which the gene was rhythmic (Rhythmic detection method, GeneCycle: *p*-value$\leq$0.01) and those for which it was not (Rhythmic detection method, GeneCycle: *p*-value>0.5). Because of the difficulty of setting reliable thresholds for rhythmicity [14], we ignored intermediate *p*-values. For each gene, we estimated the difference $\delta$ of expression levels between these two groups of tissues (see Formula 7 in Methods). As expected, genes tend to be more highly expressed in tissues where they are rhythmic (Student's test, testing the hypothesis that the $\delta$ distribution mean is equal to 0: $\delta_{mean}$ = 0.0146, t = 28.29, df = 11683, *p*-value<2.2e−16, 95% CI [0.0136, 0.0156]). We also provide results obtained from other datasets in S3 Table, although they must be taken with caution since only 2 to 4 tissues were available, and sometimes data were coming from different experiments. Of note, for proteomic data, the distributions of $\delta$ are bimodal (S3 Fig), separating rhythmic proteins into two groups, with low or high protein levels in the tissues in which they are rhythmic. A hypothesis is that for some tissue-specific proteins the rhythmic regulation is not tissue-specific, making them rhythmic also in tissues where they are lowly expressed. But the very small sample size does not allow us to test it, and we caution against any over-interpretation of this pattern before it can be confirmed.

## Rhythmic genes are tissue-specific

To clarify whether rhythmic genes tend to be tissue-specific highly expressed genes, we also analysed the relation between the number of tissues in which a gene is rhythmic and its tissue-specificity of expression $\tau$ [26, 27]. For the mouse circadian dataset (transcriptomic) with 11

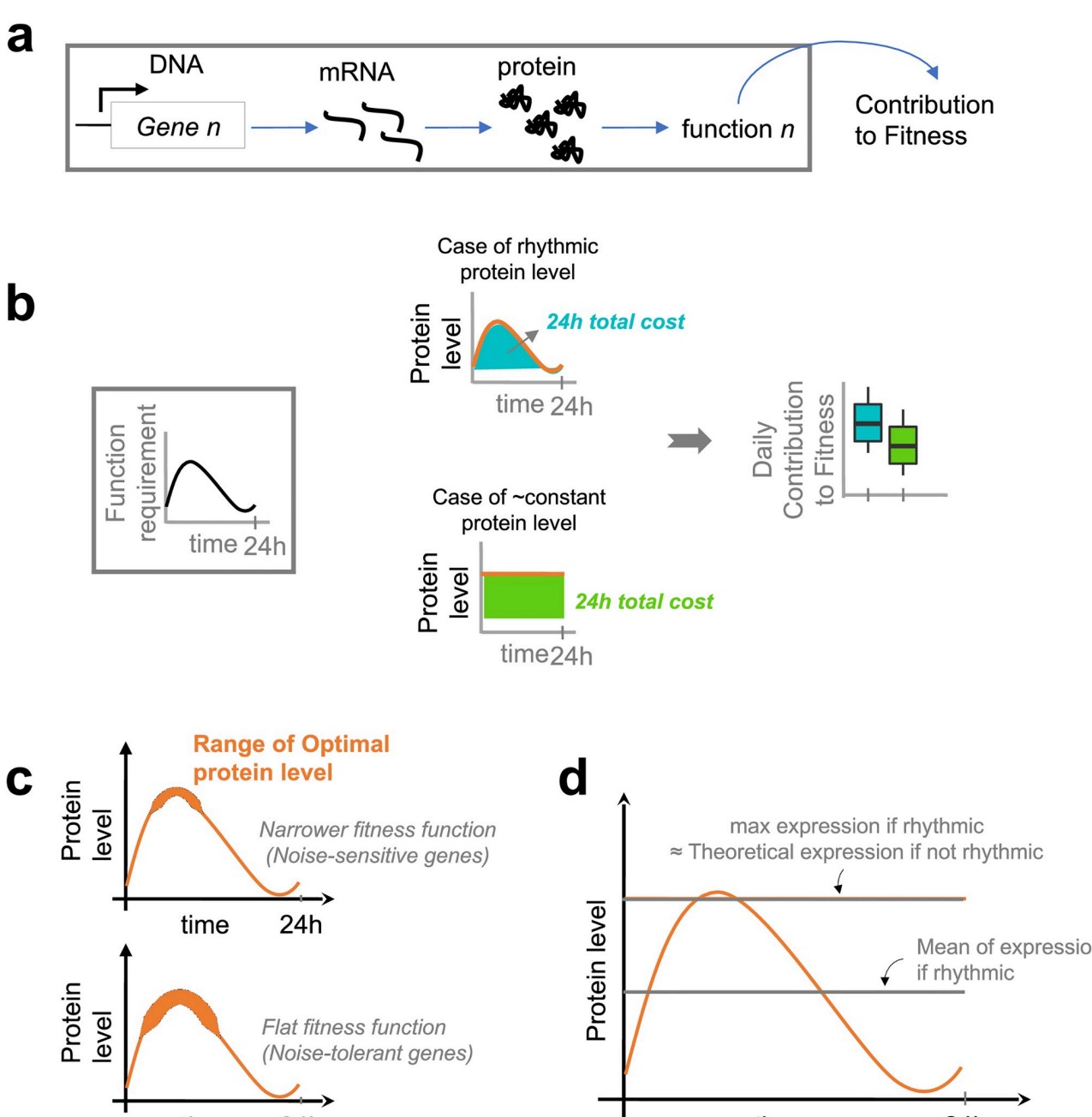

**Fig 1. a)** Gene expression contributes to organismal fitness. **b)** Rhythmic protein regulation presents a trade-off between the costs generated by integration into the rhythmic system (costs of complexity) and the advantages provided plus the costs saved over 24 hours. The middle exemplifies two extreme behaviors, while the right shows the distribution expected from populations of genes which follow these behaviors. **c)** The range of high fitness protein levels depends on the sensitivity of the function to deviations from an optimal level. We use the term "narrower" following Hausser et al. [20]. Noise sensitive genes have narrower fitness function, *i.e.* a small deviation from the optimum rapidly decreases the contribution to fitness. Precision is less important for genes with flat fitness functions. **d)** Mean or maximum expression level calculated from time-series datasets (see Methods). We assume that, in the absence of rhythmic regulation, the constant optimal level is included between the mean and the maximum expression level observed in rhythmic expression.

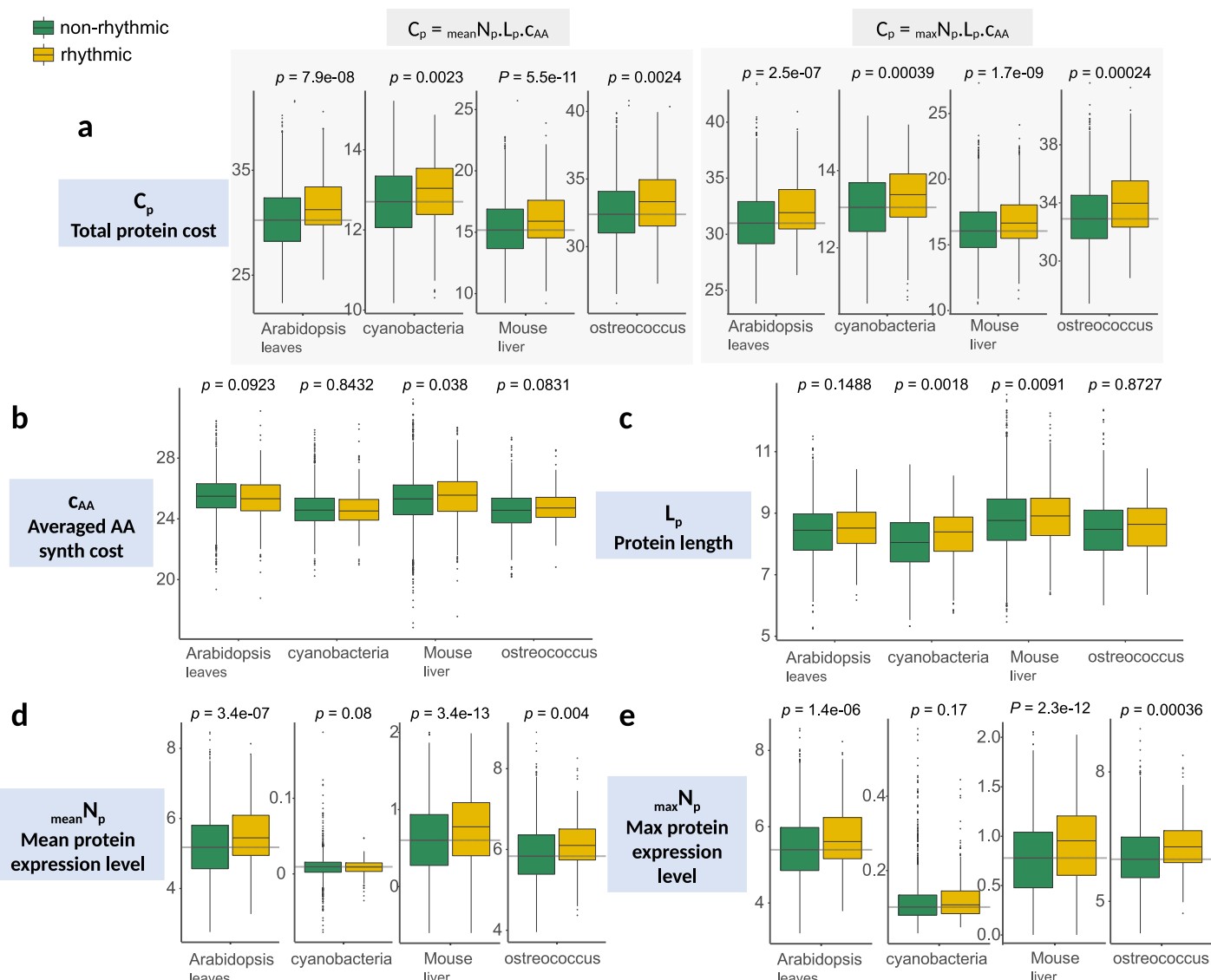

**Fig 2. Among the factors of expression costs, expression level is the main factor explaining the higher cost observed in rhythmic proteins. a)** The total cost of rhythmic proteins is higher than those of other proteins. **b)** With the exception of mouse liver, rhythmic proteins do not contain more expensive amino-acids than other proteins. **c)** Rhythmic proteins can be longer in some species. **d-e)** Mean or maximum expression level calculated from time-series datasets: rhythmic proteins are highly expressed proteins. Boxplots are log scaled except for the averaged AA synthesis cost. The first 15% of proteins from *p*-values ranking (from the most rhythmic to the most un-rhythmic genes) obtained from the rhythm detection algorithms were classified as rhythmic.

tissues, partial correlations show that genes whose rhythmicity is tissue-specific have tissue-specific expression ($\tau$, Formula 8, versus number of rhythmic tissues: Pearson's correlation = -0.37, t = -5.3e+01, $p < 2.2e^{-16}$; see S4 Table for Spearman's correlation test). Results obtained with other datasets are in S4 Table; results for the baboon dataset should be taken with caution for reasons discussed in our previous work [14]. The partial correlation seems to be stronger at the transcriptional level, although data was available in much less tissues at the pro- tein level (only mouse forebrain, cartilage, liver, and tendon). Moreover, for tissue-specific genes ($\tau$ >0.5), the signal of rhythmicity correlates with expression level over tissues (S4(a) Fig). While Spearman's correlation is clearly skewed towards negative correlations, *i.e.* lower *p*-values thus

stronger signal of rhythmicity in the tissue where genes are more expressed, Pearson's correlation also has a smaller peak of positive correlations (S4(a) Fig), suggesting a subset of genes which are less rhythmic in the tissues where they are most expressed. We show that tissue-specific genes which are mostly rhythmic in tissues where they are highly expressed are under stronger selective constraint than those which are rhythmic in tissues where they are lowly expressed (S4(b) Fig). Thus, rhythmic expression of this second set of genes might be under weaker constraints. The dominant signal overall is that in a given tissue, rhythmically expressed genes tend to be those which are tissue-specifically expressed in this tissue.

## Lower cell-to-cell variability of genes with rhythmic transcripts

Increasing transcription for a fixed amount of protein can decrease the noise in final protein levels [20], thus, genes with rhythmic mRNAs can have lower noise at their mRNA level peak. We predict that genes with rhythmic mRNAs are noise-sensitive genes (Fig 1c). To test this, we compared the noise distribution between rhythmic and non-rhythmic genes. We used $F^*$ of Barroso et al. [28] to control for the correlation between expression mean ($\mu$) and variance ($\sigma^2$). It was the most efficient compared with other methods (section 6.4 in S1 File). To evaluate expression noise, we used single-cell RNA data from *Mus musculus* liver, lung, limb muscle, heart, and aorta [29] and *Arabidopsis thaliana* roots [30] (S2 Table). These data are from a single time-point each time. In the absence of other data, we have to assume that this noise calculated at the transcriptional level is representative of noise at the protein level, or in any case functionally relevant. If anything, we expect this to provide an under-estimation of any relation between noise and rhythmicity, since noise at other time points is missed. Then, we assessed rhythmicity based on time-series datasets: RNA and proteins from Arabidopsis leaves and mouse liver from the data used above; and RNA only in mouse lung, kidney, muscle, heart, and aorta (S2 Table). Results are given in a simplified way in Table 1 for the mouse; full results, which include Arabidopsis, are provided in S5 Table. In most cases, we found lower noise for genes with rhythmic mRNA (row *a* of Table 1 and S5 Table).

We obtained the same results using less stringent cut-offs when calling genes rhythmic ($p<0.05$, results not shown). Genes with both rhythmic proteins and mRNA had lower noise (row *c* of Table 1), although differences weren't significant. Our results show that noise estimated at a single time-point is globally reduced for genes with rhythmic regulation at the transcriptional level. Since rhythmic genes are not all in the same phase (Fig 2a and S1 File), we expect this result obtained for a given time-point (average noise estimated at a single time-point from scRNA) to be general to all time-points (section 6.3 in S1 File). Assuming that genes with low noise are noise-sensitive (and thus noise is tightly controlled), these results suggest that rhythmic genes have, individually, their noise periodically and drastically reduced through periodic high accumulation of their mRNAs. In Arabidopsis, the single-cell data used are from the root, while transcriptomic time-series data used to detect rhythmicity are from the leaves, which limits the interpretation. Despite this limitation, we found no evidence of lower noise for genes that are rhythmic at the protein level (rows *b* and *e* of Table 1, and S5 Table), and trends towards lower noise in almost all cases for genes with rhythmic mRNAs (rows *a*, *c*, and *d* of Table 1).

## Genes with rhythmic transcripts are under stronger selection

Finally, we compared protein evolutionary conservation (estimated by the dN/dS ratio) between rhythmic and non-rhythmic genes. Results are given in a simplified way in Table 2; full results are provided in S6 Table. In all cases, in plants, vertebrates, and insects, we found that genes with rhythmic mRNA levels were significantly more conserved (triangles in S8

**Table 1. Simplified table showing the results of the Welch two sample t-test testing the hypothesis that the noise is equal between rhythmic versus non-rhythmic transcripts (a), or proteins (b), or between rhythmic versus non-rhythmic transcripts among rhythmic proteins (c) or among non-rhythmic proteins (d), and between rhythmic versus non-rhythmic proteins among genes with constant transcripts (e).** $F^*$ is an estimation of the noise based on Barroso et al. method [28]. Complete results are provided in S5 Table.

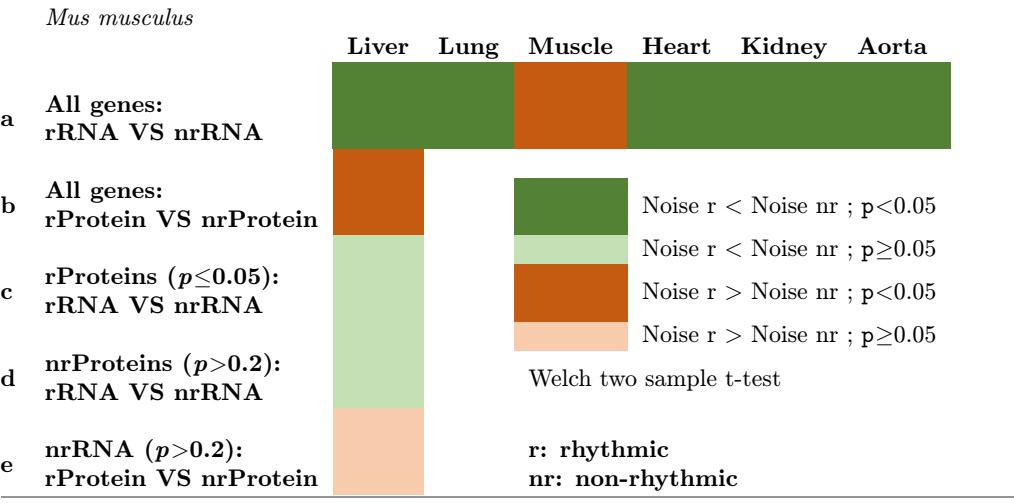

Table). We obtained similar results after controlling for gene expression bias (Table 2 and background boxes in S8 Table), since higher expressed genes are known to be more conserved genes. Interestingly, we didn't find a clear tendency for higher conservation in rhythmic proteins (Table 2). In Arabidopsis, genes rhythmic at both levels were less conserved than genes rhythmic at only one level (Table 2), and results were unclear for mouse. These results suggest that rhythmic expression at the transcriptional level plays an important role for genes under strong purifying selection, consistent with the results for tissue-specific rhythmic genes.

## Discussion

In this work, we investigate the evolutionary origins of rhythms in gene expression that respond to factors oscillating on a 24-hour time scale. This scale is longer than those of many biological oscillators at the molecular level, which are ultradian; for instance $\sim$3h for the p53-Mdm2 system, or on the order of minutes to seconds for calcium oscillations. Hence, natural selection was probably instrumental in reaching the longer nycthemeral period of gene expression for many genes.

The endogenous generation of circadian rhythms is an anticipation strategy, which is optimized if the internal clock resonates with the external cycle [31, 32]. The autonomous nature of such mechanisms provides a clear advantage to the organism able to anticipate its environmental changes before they take place, allowing it to be "ready" before organisms who would not be endowed with such capacity. In terms of adaptability in a given environment, if we consider nycthemeral rhythms without considering their endogenous or exogenous nature, one can ask the question of the evolutionary origin of maintaining large cyclic biological systems, with gene and tissue specific regulation. Indeed, they are very widely found among living organisms, at every level of biological systems, highlighting their necessity for adapted phenotypes.

**Table 2. Simplified table showing the results of the Welch two sample t-test testing the hypothesis that dN/dS ratio is equal between rhythmic versus non-rhythmic transcripts (a), or proteins (b), or between rhythmic versus non-rhythmic transcripts among rhythmic proteins (c), and between rhythmic versus non-rhythmic proteins among genes with rhythmic transcripts (d).** In S8 Table, triangles give the result of the Welch two sample t-test without controlling for the effect of gene expression level. Complete results are provided in S6 Table.

| | | | All genes: rRNA vs. nrRNA | All genes: rProtein vs. nrProtein | rProteins ($p \leq 0.05$): rRNA vs. nrRNA | rRNA ($p \leq 0.01$): rProtein vs. nrProtein |
|---|---|---|---|---|---|---|
| PLANT | **Arabidopsis** | **Leaves** | light green | peach | dark orange | peach |
| VERTEBRATES | **Mouse** | **Liver** | dark green | peach | dark green | light green |
| | | **Lung** | light green | | | |
| | | **Muscle** | peach | | | |
| | | **Heart** | dark green | | | |
| | | **Kidney** | dark green | | | |
| | | **Aorta** | dark green | | | |
| | | **Tendon** | | peach | | |
| | | **Forebrain** | | light green | | |
| | | **Cartilage** | | dark green | | |
| | **Rat** | **Lung** | light green | | | |
| INSECTS | **Anopheles** | **Head** | dark green | | | |
| | | **Body** | light green | | | |
| | **Aedes** | **Head** | light green | | | |

- ■ (dark green) dN/dS of rhythmic genes < dN/dS of non-rhythmic genes; $p < 0.05$
- ■ (light green) dN/dS of rhythmic genes < dN/dS of non-rhythmic genes; $p \geq 0.05$
- ■ (dark orange) dN/dS of rhythmic genes > dN/dS of non-rhythmic genes; $p < 0.05$
- ■ (peach) dN/dS of rhythmic genes > dN/dS of non-rhythmic genes; $p \geq 0.05$

An important question which we have addressed is why some genes are rhythmic at the RNA but not the protein level, and vice versa. Apart from mechanistic causes that explain how, it was not clear why in some cases the rhythmicity is initiated early then lost over the processes (from transcription initiation to protein decay), while in others it is initiated later. In some cases, these lost or gained rhythms might be by-products of the evolution of regulatory processes, but it seems unlikely that all nycthemeral genes be due only to genetic drift. Our previous results showing that rhythmicity of gene expression tends to be a conserved property [14] support this perspective. Indeed, we have found a stronger conservation signal of rhythmic expression for evolutionarily closer species [14]. Moreover, a reasonable assumption is that rhythmic expression can have costs, *e.g.*, of regulation, and thus it is not always obvious that it should be adaptive. For instance, regulatory dynamics can cause substantial changes in noise levels, *e.g.* the noise strength immediately following gene induction is almost twice the final steady-state value [19].

## Rhythmic proteins: Reduction of overall costs

Our calculation of energetic costs is based on the costs that the cell needs to produce a given steady state protein level. The cell also consumes energy to maintain the steady state protein level. We argue that these costs should correlate with our estimation of expression costs (see section 3 in S1 File). Indeed, the costs of maintaining a steady state protein level are higher for highly expressed than for lowly expressed genes, even if higher expressed proteins have longer half-lives (see section 4 in S1 File). Thus, even taking into account a more complex cost calculation would not change our observation that rhythmic genes are costlier.

Our results suggest that rhythmicity of protein expression has been favored by selection for cost control of gene expression, while keeping sufficient expression levels. In the case of rhythmic genes, what would this sufficient constant level be? We can propose two hypotheses. The first is that it would be the mean expression over the period, since this maintains the same overall amount of protein. The second is that it would be the maximum over the rhythm period, since that is the level needed at least at some point. The second hypothesis explains better the existence of this maximum level during the cycle. Of note, it also strengthens the case for selection on expression cost. Our results also suggest that rhythmicity at the protein level only minimizes costs within the constraints of the rhythmic proteins, without making these costs similar or smaller than those of constant proteins. They also suggest that rhythmic regulation of transcription of highly expressed genes does impact the energy expenditure, consistent with Cheng et al. [33], but as a by-product of the selection acting at protein level. Finally, rhythmic regulation of highly expressed genes might be driven by other costs than energy, *i.e.* high expression levels could be detrimental, *e.g.* because of toxicity. For instance, by altering protein-protein interactions, generating undesirable enzyme activities, or by antagonistic pleiotropy of genes. Thus, for rhythmic genes, the constant level should at least correspond to the mean expression level (Fig 1d). We provide results obtained using both the maximum and the mean of expression in Fig 2a.

Another hypothesis is that rhythmicity at protein level could have been selected for noise adjustment. If that were the case, we would expect that for a given mRNA level, the peak level of proteins (*i.e.*, the highest translation rate) would be associated with lower noise to increase the precision when the protein level is highest. However, both theoretical and empirical arguments contradict this expectation. For a given mRNA level, theoretical models predict that protein level will be negatively correlated with noise [20, 34] (section 6.2 in S1 File). Moreover, measurements of cell-to-cell variability of protein level show that for a given mRNA level, changes in protein level have little to no impact on gene expression precision [20] (section 6.2 in S1 File). Thus, translational efficiency does not seem to be the main driver of expression noise, and selection on noise can- not explain rhythmicity at protein level.

## Rhythmic mRNAs: Noise reduction

Transcription rather than translation seems to be the main source of overall expression noise (section 6.1 in S1 File). However, we can only conclude regarding the transcriptional noise. Small fluctuations of expression away from the optimal wild-type expression have been shown to impact organismal fitness in yeast, where noise is nearly as detrimental as sustained (mean) deviation [35]. Furthermore, noise-increasing mutations in endogenous promoters have been found to be under purifying selection [36], and noise has been shown to be under selection in fly development [21]. In this study, we found lower noise at least at a single time-point among rhythmic transcripts, compared to constant genes. Rhythmic expression of RNAs might be a way to periodically reduce expression noise of highly expressed genes (Fig 2 and S1 Fig), which are under stronger selection. Indeed, we found that genes with rhythmic transcripts are under stronger selection, even controlling for expression level effect. As proposed by Horvath et al. (2019) and supported by results in mouse by Barroso et al. (2018), genes under strong selection could be less tolerant to expression noise. Thus, periodic accumulation of mRNAs might be a way to periodically reduce transcriptional noise of noise-sensitive genes (Fig 1c), *i.e.* genes under stronger selection. However, our results are limited by the fact that noise estimation is based on a single time-point measurement since no scRNA time-series data are currently available for these species. Since the peak time of rhythmic transcripts is distributed across all times (Fig 2a in S1 File), the mean noise estimated at a given time-point includes the noise of the

genes that are peaking at that time (lowest noise) and all the others that have a higher noise than those at their own peak time-point (Fig 2b in S1 File). Thus, genes peaking at the time-point of the scRNA measurement had sufficiently low noise to produce an averaged noise in rhythmic genes much lower than constant genes. We suggest that, over a gene-specific threshold of mRNA level, the likelihood that ribosomes interact with mRNA molecules increases much faster than under this threshold, allowing to maximize expression accuracy at the peaks. Future experiments would allow to test this suggestion.

We are aware that this theory is difficult to test experimentally. An interesting direction would be the measurement the cell-to-cell protein fluorescence at different time-point for different tissues and species, providing real-time information on the noise fluctuations at various levels. Moreover, the fitness effect of noise could be estimated from growth rates in unicellular organisms measured at different time-point, under conditions which favor rhythmicity or which don't.

Stochasticity of gene expression could also allow to maintain an oscillator system that would be damped otherwise, according to deterministic simulations [37]. Furthermore, expression noise could generate phenotypic diversity, and improve fitness in fluctuating environments [38–43]. Thus, rhythmicity of transcripts might lead to periodic stochasticity that maintains the amplitude of oscillations as well as lead to alternative molecular phenotypes.

Higher precision allows to increase the robustness of gene expression when the function is most needed, while higher stochasticity allows to present diverse molecular phenotypes. Thus alternating between these two states might improve the fitness of organisms in fluctuating environments. We also found that rhythmic expression involves highly expressed, tissue-specific genes, consistent with the observation of tissue-specific rhythms. Selection for rhythmic regulation of highly expressed genes might be due to an efficient reduction of periodic protein-protein misinteractions. Finally, since genes under strong selection could also become less tolerant to high expression noise levels, our results suggest that rhythmicity at the mRNA level might have been under strong selection for these noise-sensitive genes.

## Materials and methods

### Datasets

Details of the datasets are available in S2 Table.

**Mus musculus.**   Mouse liver transcriptomic and proteomic time-series datasets come from Mauvoisin et al. [17]. Original protein counts dataset was downloaded from ProteomeXchange (PXD001211) file Combined_WT, and cleaned from data with multiple Uniprot.IDs affectations between raw peptides data. Time-serie transcriptomic data was downloaded from the National Center for Biotechnology Information (NCBI) Gene Expression Omnibus (GEO) accession (GSE33726) [44] Multi-tissues time-serie transcriptomic data used for tissue-specificity expression comparisons are microarray data from Zhang et al. [9] downloaded from the NCBI GEO accession GSE54652 (see Materials of Laloum and Robinson-Rechavi [14] for more details). Tissues analysed are: adrenal gland, aorta, brain stem, brown adipose, cerebellum, heart, kidney, liver, lung, muscle, and white adipose. We have volontarily excluded the hypothalamus since it contains the Supra-Chiasmatic Nucleus known to have core clock genes that synchronise the body. Therefore, their rhythmicity is probably dominated by other causes than a regulation of cost.

Multi-tissues time-serie proteomic data come from: Mauvoisin et al. [17] for the liver, Noya et al. [45] for the forebrain, Chang et al. [46] for the tendon, and Dudek et al. [47] for the cartilage. Finally, single-cell data of thousand of cells in organs for which we had time-series datasets, *i.e.* liver, lung, kidney, muscle, aorta, and heart, were downloaded from figshare using R

objects from FACS single-cell datasets [29]. We kept cellular sub-types which were found in at least 100 cells and that we considered to be characteristic of the tissue. This leads to keep: `endothelial cell of hepatic sinusoid` and `hepatocyte` for the liver; `epithelial cell of lung`, `lung endothelial cell`, and `stromal cell` for the lung; `endothelial cell`, `mesenchymal stem cell`, and `skeletal muscle satellite cell` for the muscle; `cardiac muscle cell`, `endocardial cell`, and `endothelial cell` for the heart; `endothelial cell` for the aorta; `endothelial cell`, `epithelial cell of proximal tubule`, and `kidney collecting duct epithelial cell` for the kidney.

**Arabidopsis thaliana.** Leaves time-series proteomic data are the Dataset I from Krahmer et al. [48], cleaned from data with multiple protein identifications. Leaves time-series transcriptomic data were downloaded from the NCBI GEO accession (GSE3416) [49]. Single-cell data of twenty root cells were downloaded from the NCBI GEO accession (GSE46226) [30].

**Ostreococcus tauri.** Unicellular alga proteomics time-series dataset (normalized abundances) come from Noordally et al. [50]. Transcriptomic time-series dataset come from Monnier et al. [51], was downloaded from the NCBI GEO accession (GSE16422) and was cleaned for genes with too much missing values (more than seven).

**Synechococcus elongatus (PCC 7942).** Unicellular cyanobacterium proteomics time-series dataset come from Guerreiro et al. [52]. Transcriptomic time-series dataset come from Ito et al. [53] and was recovered from Guerreiro et al. [52].

**Drosophila melanogaster.** Transcriptomic time-series datasets for the body, the head, and the heart, come from Gill et al. [54] and were downloaded from the NCBI GEO accession (GSE64108) (see Materials of Laloum and Robinson-Rechavi [14] for more details).

**Papio anubis [Olive baboon].** Multi-tissues time-series transcriptomic data used for tissue-specificity expression comparisons are RNA-seq data from Mure et al. [55] (see Materials of Laloum and Robinson-Rechavi [14] for more details).

## Pre-processing

For each time-series dataset, only protein coding genes were kept. ProbIDs assigned to several proteins were removed. Probesets were cross-referenced to best-matching gene symbols by using either Ensembl BioMart software [56], or UniProt [57].

## Rhythm detection

To increase power of rhythm detection [14], we considered biological replicates as new cycles when it was possible. We used `GeneCycle` R package (version 1.1.4) [58] available from CRAN and used the robust.spectrum function developped by [59]—with parameters *periodicity.time = 24* and *algorithm = "regression"*—that computes a robust rank-based estimate of the periodogram/correlogram and that we improved with the `try-catch` function to avoid error of dimension with MM-estimation method. When the *p*-values distribution obtained did not correspond to the expected distribution—skewed towards low *p*-values because of the presence of rhythmic genes—we used the results obtained by the rhythm detection method used by the original paper from where the data came after checking they presented a classic skewed *p*-values distribution. Finally, for each gene or protein having several data (ProbIDs or transcripts), we combined *p*-values by Brown's method using the `EmpiricalBrowns-Method` R package (S1 File). Thus, for each dataset, we obtained a unique rhythm *p*-value per gene or per protein. Due either to low-amplitude or less-accurate measurements, it was more challenging to identify rhythms in proteomics data. That is why, in general, we used lower stringency for proteins.

## Consistent gene expression levels

Tissue-specific mRNA or protein abundances were the average or the maximum level of the $n$ time-points such as for gene $i$:

$$_{max}N_i = \frac{_{max1}N_{i,j} + _{max2}N_{i,j}}{2} \quad \text{with} \quad _{max1}N_{i,j} \neq _{max2}N_{i,j} \quad 0 \leq j \leq n \tag{2}$$

$$_{mean}N_i = \frac{\sum_{j=1}^{n} N_{i,j}}{n} \quad \text{with} \quad n: \text{ the number of time} - \text{points} \tag{3}$$

$N_{pi}$: for the abundance of the protein i
$N_{RNAi}$: for the abundance of the transcript i
In mouse, we have used peptide level data to compare protein abundances in the calculation of expression costs.

## Expression costs

Energetic costs of each AA (unit: high-energy phosphate bonds per molecule) come from Akashi and Gojobori [60], or Wagner [15] which are linearly correlated (S5 Fig). The averaged AA synthesis cost of one protein of lenght $Lp$ is:

$$\bar{c}_{AA} = \frac{\sum_{j=1}^{L_p} c_{AAj}}{L_p} \tag{4}$$

Protein sequences come from FASTA files downloaded from EnsemblPlants [56] or UniProt [57] for: Proteome UP000002717 for *Synechococcus elongatus* PCC 7942, Uniprot Reviewed [Swiss-Prot] for *Mus musculus*, Proteome UP000009170 for *Ostreococcus tauri*.

Main results use Wagner [15] AA costs data. We provide supplementary results using Akashi and Gojobori [60] AA costs data (S6 Fig).

## Multi-tissues analysis

To obtain comparable expressions levels between different tissues or datasets, we normalized expression values by Z-score transformation such as in the dataset of $n$ genes, the mean expression of the gene $i$ becomes:

$$_{mean}Z_i = \frac{_{mean}N_i - \bar{N}}{_{max}Z.\sigma} \quad \text{with} \quad _{mean}N_i: \text{ the average expression level of gene } i \text{ (formula (3))}$$

$$\bar{N} = \frac{\sum_{i=1}^{n} N_i}{n}$$

$\sigma$: the standard deviation of $_{mean}N_i$

$_{max}Z$: the maximal value of the $_{mean}Z-$scores

$n$: the number of genes

$$(5)$$

and the maximal expression of the gene $i$ becomes:

$$_{max}Z_i = \frac{_{max}N_i - \bar{N}}{_{max}Z.\sigma} \quad \text{with} \quad$$

$_{max}N_i$: the maximal expression level of gene $i$ (formula (2))

$_{max}Z$: the maximal value of the $_{max}Z-$scores $\qquad$ (6)

$\sigma$: the standard deviation of $_{max}N_i$

$$Z_i \in [0, 1]$$

To compare the expression levels between the set of $n_r$ rhythmic tissues and the set of $n_{\bar{r}}$ non-rhythmic tissues, we estimated the difference ($\delta$) of expression levels between these two groups for each gene $i$ such as:

$$\delta_i = \frac{\sum_{j_r=1}^{n_r} \left(_{mean}Z_{i,j_r}\right)}{n_r} - \frac{\sum_{j_{\bar{r}}=1}^{n_{\bar{r}}} \left(_{mean}Z_{i,j_{\bar{r}}}\right)}{n_{\bar{r}}} \qquad (7)$$

with $\quad n_r$: the number of tissues in which gene $i$ is rhythmic ($p \leq 0.01$ or $0.05$)

$n_{\bar{r}}$: the number of tissues in which gene $i$ is not rhythmic ($p > 0.1$ or $0.5$)

$_{mean}Z_{i,j_r}$: the mean expression level $Z -$ score normalized of gene $i$ in

$\qquad$ the tissue $j_r$ in which it is rhythmic

$_{mean}Z_{i,j_{\bar{r}}}$: idem for non $-$ rhythmic tissue

$_{mean}Z_i \in [0, 1]$, defined as formula (5)

Finally, we analysed the distribution of $\delta_i$ and generated a Student's t-test to compare its mean distribution with an expected theoretical mean of 0.

## Tissue-specificity of gene expression

To calculate a tissue-specificity $\tau$ for each gene $i$, we log-transformed the averaged gene expression and followed Kryuchkova-Mostacci and Robinson-Rechavi [27] to make expression values manageable. Thus, among the $n$ tissues, the tissue-specificity of gene $i$ is:

$$\tau_i = \frac{\sum_{j=1}^{n}(1 - \hat{N}_{i,j})}{n-1} \quad \text{with} \quad$$

$n$: the number of tissues

$$\hat{N}_{i,j} = \frac{log(_{mean}N_{i,j})}{\max_{1 \leq j \leq n}(log(_{mean}N_{i,j}))}$$

$\max\limits_{1 \leq j \leq n}(log(_{mean}N_{i,j}))$ is the maximal expression level of gene $i$ $\quad$ (8)

$\qquad$ among the $n$ tissues

$\hat{N}_{i,j} \in [0, 1]$

This tissue-specificity formula was described by Yanai et al. [26]. Finally, we performed linear regressions to analyse the respective and the interaction influences of gene expression level and tissue-specificity into the rhythmicity. For mouse [9], we used RNA-seq data to estimate the mean expression used for the calculation of $\tau$, and used rhythm $p$-values obtained from

microarray dataset since there was more time-points in the microarray time-series (as discussed in benchmark paper). S7 Fig shows the distributions of $\tau$ obtained.

## Gene expression noise quantification

We estimated a unique expression noise per gene by taking advantages of *Arabidopsis thaliana* roots [30] and Mus musculus liver, lung, kidney, muscle, aorta, and heart [29] (from figshare using R objects of FACS single-cell datasets) single-cell RNAseq data (see Materials). For Arabidopsis, we obtained gene expression for twenty root quiescent centre (QC) cells for which a total of 14,084 genes had non-zero expression in at least one of the 20 single cells. For mouse, we obtained gene expression for a number of cells ranging from 180 to 3772 cells, for which a total of genes with non-zero expression was around 19,000 genes. As in Barroso et al. [28], we kept only genes whose expression level satisfied log[FPKM + 1] >1.5 in at least one single cell. We calculated the stochastic gene expression ($F^*$) defined by Barroso et al. [28] as a measure for gene expression noise (cell-to-cell variability) designed to control the biases associated with the correlation between the expression mean ($\mu$) and the variance ($\sigma^2$) by using the lowest degree polynomial regression which decorrelate them. We tested different degrees of the polynomial regression to estimate $\log(\sigma^2)$ in the calculation of $F^*$ and measured the correlation based on Kendall's rank and linear regression slope (see section 6.4 in S1 File). The $F^*$ polynomial degree of Barroso et al. (2018) was the best method because it removed the correlation between mean expression and $F^*$ (smallest linear regression slope and Kendall's) and keep a large range of noise values for every mean expression (smallest R2) (section 6.4 in S1 File). Finally, since the peak times of rhythmic genes is largely distributed (Fig 2a in S1 File), we expect the mean noise of a given time-point to be general to all time-points (Fig 2b in S1 File), therefore giving a good indicator to estimate its biologically relevant.

## dN/dS analysis

dN/dS data have been downloaded from Ensembl BioMart Archive 99 [56]. The homologous species used are: *Mus musculus—Rattus norvegicus*; *Arabidopsis thaliana—Arabidopsis lyrata*; and *Anopheles gambiae—Aedes aegyti*. We first tested the hypothesis that rhythmic (*p*-value≤cutoff.1) versus non-rhythmic genes (*p*-value>cutoff.2) have equal ratio of non-synonymous to synonymous substitutions. Then, since rhythmic genes are now known to be enriched in highly expressed genes and because highly expressed genes are under purifying selection, we controlled for the effect of gene expression on dN/dS ratio by testing the hypothesis that rhythmic (*p*-value≤cutoff.1) versus non-rhythmic (*p*-value>cutoff.2) genes among residuals of the linear regression fitting gene expression level and dN/dS ratio. Complete results are given in S6 Table and in a simplified way in Table 2.

In R code it gives:

```
> lmTest = lm(log(dNdS) ~ log(gene.expr.level), data)
> rhythmic.residuals = lmTest$residuals[data$rhythm.
pvalue≤cutoff.1]
> nonrhythmic.residuals = lmTest$residuals[data$rhythm.
pvalue>cutoff.2]
> t.test(rhythmic.residuals, nonrhythmic.residuals)
```

Plots have been generated using ggplot2 package (version 3.3.2) run in R version 4.0.2.

## Supporting information

**S1 Fig. Mean or maximum mRNA expression level calculated from time-series datasets.**
Rhythmic transcripts are highly expressed transcripts.
(TIFF)

**S2 Fig. Linear regression analysis between the order of the rhythmicity signal (from the most rhythmic to the most un-rhythmic genes) and the total expression cost (Cp, calculated from the mean expression level).**
(TIFF)

**S3 Fig. Distribution of the difference of expression levels ($\delta$) between rhythmic tissues group and non-rhythmic tissues group obtained for every gene ($\delta$ calculated from mean expression levels; see Methods).** $\delta$ shows a bimodal distribution at protein level.
(TIFF)

**S4 Fig.** a) Histograms of Pearson's and Spearman's coefficients testing the correlation between the expression level and the rhythmicicity signal (rhythm p-value) measured across the body for every tissue-specific genes ($\tau > 0.5$). b) Student's t-test comparing the mean of dN/dS between gene sets A and B: Tissue-specific genes which are mostly rhythmic in tissues where they are highly expressed are under stronger selective constraint than those which are rhythmic in tissues where they are lowly expressed.
(TIFF)

**S5 Fig. Linear relationships of amino acid (AA) biosynthesis costs estimated by Akashi and Gojobori [60], and Wagner [15].**
(EPS)

**S6 Fig. Comparison of the averaged AA synthesis costs calculated in both groups: Rhythmic VS random genes group, using Akashi and Gojobori versus Wagner AA costs data.**
(EPS)

**S7 Fig. Histograms of tissue-specificity $\tau$.**
(TIFF)

**S1 Table. Energetic costs estimated for each amino acid.**
(XLSX)

**S2 Table. Protein and RNA time-series datasets & single-cell RNA datasets.**
(XLSX)

**S3 Table. Comparison of expression level of genes in tissues in which they are rhythmic or not.**
(XLSX)

**S4 Table. Pearson and Spearman correlation test: $\tau$ vs. the number of tissues in which the gene is rhythmic.**
(XLSX)

**S5 Table. Comparison of expression noise level (F*) between rhythmic versus non-rhythmic genes: Welch two sample t-test.**
(XLSX)

**S6 Table. Student's test testing the hypothesis that the dN/dS distributions are equal among rhythmic versus non-rhythmic genes.**
(XLSX)

**S7 Table. Linear regression slope, R2, and Kendall's correlation test of the relationships between gene expression level and variance, standard deviation (sd), or different noise estimation methods.**
(XLSX)

**S8 Table. Welch two sample t-test comparing protein evolutionary conservation (estimated by the dN/dS ratio) between rhythmic and non-rhythmic genes with, or without controlling for the effect of gene expression level (triangles).**
(XLSX)

**S1 File. Complementary information.**
(PDF)

## Acknowledgments

We thank Johanna Krahmer (CIG, UNIL, Lausanne Switzerland) for her helpful advice on datasets in plants. We also thank Qingyao Huang (DMLS, UZH, Zürich, Switzerland) and Manfredo Quadroni (PAF, UNIL, Lausanne, Switzerland) for their helpful advice on interpretation of proteomics data.

## Author Contributions

**Conceptualization:** David Laloum, Marc Robinson-Rechavi.

**Data curation:** David Laloum.

**Formal analysis:** David Laloum.

**Funding acquisition:** Marc Robinson-Rechavi.

**Investigation:** David Laloum, Marc Robinson-Rechavi.

**Supervision:** Marc Robinson-Rechavi.

**Visualization:** David Laloum.

**Writing – original draft:** David Laloum.

**Writing – review & editing:** David Laloum, Marc Robinson-Rechavi.

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
