## [Decision Letter · Decision Letter 0]

9 Mar 2022

Dear Prof. Robinson-Rechavi,

Thank you very much for submitting your manuscript "Rhythmicity is linked to expression cost at the protein level but to expression precision at the mRNA level" for consideration at PLOS Computational Biology.

As with all papers reviewed by the journal, your manuscript was reviewed by members of the editorial board and by several independent reviewers. In light of the reviews (below this email), we would like to invite the resubmission of a significantly-revised version that takes into account the reviewers' comments.

Generally I think this topic is interesting and very important to the circadian field. However, there are some substantial concerns that need to be addressed before its publication.

We cannot make any decision about publication until we have seen the revised manuscript and your response to the reviewers' comments. Your revised manuscript is also likely to be sent to reviewers for further evaluation.

Sincerely,

Guang-Zhong Wang

Guest Editor

PLOS Computational Biology

Ville Mustonen

Deputy Editor

PLOS Computational Biology

Generally I think this topic is interesting and very important to the circadian field. However, there are some substantial concerns that need to be addressed before its publication.

Reviewer's Responses to Questions

**Comments to the Authors:**

Reviewer #1: Please see the attached.

Reviewer #2: The author compared the expression levels of rhythmic and non-rhythmic proteins, and concluded that the abundance of rhythmic proteins in arabidopsis, mouse and ostreococcus was higher than that of non-rhythmic proteins, and further calculated the energy consumption of proteins according to the protein abundance, and found that the cost of rhythmic proteins was higher than that of non-rhythmic proteins.

Major comments

1) I am not an expert in proteomics, but according to my knowledge, the proteomics data of mouse liver and cyanobateria used by the authors used SILAC and TMT for protein quantitation respectively, both of which are relatively quantitative methods. Because of the different ionization efficiencies of different peptides, it is not possible to directly compare MS signals of different peptides to determine their abundance in samples, but peaks with the same peptide can be compared in different samples (PMID: 30609196). I checked the data and code using to calculate mouse liver data from Mauvoisin et al., which was used by the author. Although the calculation code was provided, the annotation was not very clear. I think the author used Ratio H/L as the expression of protein abundance. However, Ratio H/L refers to the intensity of each peptide relative to the ratio of the same peptide in the MIX group, which cannot be used for comparison between different proteins. The current relative quantitative method that can be used for comparing proteins can be quantified by label free, processed by MaxQuant, and compared with the IBAQ value therein.

2) The authors calculated the energy expended in protein translation, but not in RNA transcription. The authors assume that the energy cost of RNA transcription is negligible compared to that of protein translation. This is possible in terms of calculating the total energy of transcription and translation, but on the other hand, transcription takes place in the nucleus, and its energy use might be thought of as being in a relatively closed system. Is it possible to calculate the energy of RNA transcription alone? Do the authors think such calculations have biological significance? If agree, I suggest that this work be supplemented.

3) In the absence of other data, the authors have to assume that this noise calculated at the transcriptional level is representative of noise at the protein level, or in any case functionally relevant. I don't think this is a reasonable estimate because the error is too large to make the protein noise calculated using this assumption meaningful. Perhaps the authors should have used other methods to estimate the amount of protein and protein noise from single cell data, for example, ribosome profiling. Or the author can discuss the error of the estimation.

Minor comments

1) To test the hypothesis that genes tend to be rhythmic in the tissues in which they are highly expressed, the authors used a mouse circadian dataset with 11 tissues (transcriptomics from Zhang et al. (2014), Supplementary Table S2). As far as I know, that dataset includes a total of 12 mouse tissues. What about the other tissues? If a selection was made, perhaps the author should state the criteria in the data section.

2) In this paper, gene expression noise is a very important concept. Although the author directly uses the calculation method by Barroso et al. (2018) when quantifying noise expression, I think it is still necessary to explain the quantification process of noise and the biological significance of noise in more detail in the method section.

**Have the authors made all data and (if applicable) computational code underlying the findings in their manuscript fully available?**

Reviewer #1: Yes

PLOS authors have the option to publish the peer review history of their article (what does this mean?). If published, this will include your full peer review and any attached files.

Reviewer #1: No

Reviewer #2: No
---

## [Decision Letter · Decision Letter 1]

27 May 2022

Dear Prof. Robinson-Rechavi,

Thank you very much for submitting your manuscript "Rhythmicity is linked to expression cost at the protein level but to expression precision at the mRNA level" for consideration at PLOS Computational Biology. As with all papers reviewed by the journal, your manuscript was reviewed by members of the editorial board and by several independent reviewers. The reviewers appreciated the attention to an important topic. Based on the reviews, we are likely to accept this manuscript for publication, providing that you modify the manuscript according to the review recommendations.

The second reviewer still has some points need to be carefully addressed.

Sincerely,

Guang-Zhong Wang

Guest Editor

PLOS Computational Biology

Ville Mustonen

Deputy Editor

PLOS Computational Biology

[LINK]

The second reviewer still has some points need to be carefully addressed.

Reviewer's Responses to Questions

**Comments to the Authors:**

Reviewer #1: Please see attached.

Reviewer #2: I am glad that the authors have given detailed replies to my previous comments.

In my last review, I mentioned that the abundance of different proteins obtained by previous quantitative methods cannot be compared with each other. The authors reanalyzed it in this revision. It's nice to see that the new results are slightly different from the previous results that there is no change in the conclusions. However, the authors’ reply is not very clear about how to improve the method. I think the authors have moved from analyzing the abundance at the protein level to the peptide level. Is my understanding correct?

As for the second question I mentioned earlier, “whether it is necessary to calculate mRNA-related expression energy”, the authors calculated that the energy required to produce protein was more than 160 times that of mRNA, and the energy cost of protein was much higher than that of mRNA. Therefore, it makes sense to consider just the protein without needing to be very elaborate. I think the explanation given by the author is reasonable.

As for another comment, that transcriptional noise is not representative of translation noise, I would like to thank the authors for providing relevant article on the fact that transcriptional noise is a major source of noise in bacteria. This substitution makes sense in bacteria and yeast. Unfortunately, there is no data on multicellular species, which would make this paper more rigorous.

The author has answered the reasons why the hypothalamus was not included in the analysis. I think the reasons given are reasonable.

In addition, I have a new suggestion for the author to consider.

Major comments

1)As briefly mentioned in the manuscript, other factors such as ‪protein-protein misinteractions, undesirable enzyme activities or antagonistic pleiotropy may also explain the rhythmicity of gene transcription. The authors should provide strong evidences to exclude the possibility that those factors can lead to noise reduction in ‪rhythmic mRNAs.

And no more other comments.

**Have the authors made all data and (if applicable) computational code underlying the findings in their manuscript fully available?**

Reviewer #1: None

Reviewer #2: None

PLOS authors have the option to publish the peer review history of their article (what does this mean?). If published, this will include your full peer review and any attached files.

Reviewer #1: No

Reviewer #2: No

Figure Files:

Data Requirements:

Reproducibility:

References:

---

## [Editor Report · Decision Letter 2]

17 Jul 2022

Dear Prof. Robinson-Rechavi,

We are pleased to inform you that your manuscript 'Rhythmicity is linked to expression cost at the protein level but to expression precision at the mRNA level' has been provisionally accepted for publication in PLOS Computational Biology.

Best regards,

Guang-Zhong Wang

Guest Editor

PLOS Computational Biology

Ville Mustonen

Deputy Editor

PLOS Computational Biology

---

## [Editor Report · Acceptance letter]

5 Sep 2022

PCOMPBIOL-D-21-02216R2 

Rhythmicity is linked to expression cost at the protein level but to expression precision at the mRNA level

Dear Dr Robinson-Rechavi,

I am pleased to inform you that your manuscript has been formally accepted for publication in PLOS Computational Biology. Your manuscript is now with our production department and you will be notified of the publication date in due course.

With kind regards,

Olena Szabo
